# Application of Machine Learning Methods for Pallet Loading Problem

**Batin Latif Aylak** [1,*]**, Murat İnce** [2]**, Okan Oral** [3]**, Gürsel Süer** [4]**, Najat Almasarwah** [5]**, Manjeet Singh** [6]
**and Bashir Salah** [7]

1. Department of Industrial Engineering, Turkish-German University, Şahinkaya Caddesi 106, Beykoz, Istanbul 34820, Turkey
2. Vocational School of Technical Science, Isparta University of Applied Sciences, Isparta 32260, Turkey; muratince@isparta.edu.tr
3. Department of Mechatronics Engineering, Akdeniz University, Antalya 07058, Turkey; okan@akdeniz.edu.tr
4. Stocker Center 283, Industrial and Systems Engineering, Ohio University, Athens, OH 45701, USA; suer@ohio.edu
5. Department of Industrial Systems Engineering, Mutah University, Alkarak 61710, Jordan; najat.eid@mutah.edu.jo
6. DHL Supply Chain, Solutions Design, Westerville, OH 43082, USA; Manjeet.Singh3@dhl.com
7. Department of Industrial Engineering, College of Engineering, King Saud University, P.O. Box 800, Riyadh 11421, Saudi Arabia; bsalah@ksu.edu.sa
* Correspondence: batin.latif@tau.edu.tr

**Abstract:** Because of continuous competition in the corporate industrial sector, numerous companies are always looking for strategies to ensure timely product delivery to survive against their competitors. For this reason, logistics play a significant role in the warehousing, shipments, and transportation of the products. Therefore, the high utilization of resources can improve the profit margins and reduce unnecessary storage or shipping costs. One significant issue in shipments is the Pallet Loading Problem (PLP) which can generally be solved by seeking to maximize the total number of boxes to be loaded on a pallet. In many previous studies, various solutions for the PLP have been suggested in the context of logistics and shipment delivery systems. In this paper, a novel two-phase approach is presented by utilizing a number of Machine Learning (ML) models to tackle the PLP. The dataset utilized in this study was obtained from the DHL supply chain system. According to the training and testing of various ML models, our results show that a very high (>85%) Pallet Utilization Volume (PUV) was obtained, and an accuracy of >89% was determined to predict an accurate loading arrangement of boxes on a suitable pallet. Furthermore, a comprehensive analysis of all the results on the basis of a comparison of several ML models is provided in order to show the efficacy of the proposed methodology.

**Keywords:** logistics; machine learning; pallet loading problem (PLP); classifiers

## 1. Introduction

Because of the advancements in transportation technology, a large number of products are shipped to the customers globally using a wide variety of transport vehicles such as trucks, planes, ships, etc., on a daily basis. These products are first packed in boxes and placed on pallets and then loaded into trucks, containers, and other transportation options [1]. This whole process necessitates optimizing the smallest volume of the utilization of resources at each step in a cost-effective manner. For instance, an efficient and optimized procedure of pallets loading can lead to a significant amount of cost savings. Hence, it is quite evident that efficient utilization of pallets may entail the reduction of goods traffic, thereby preserving the company's time, resources, and the involved costs. Therefore, an efficient solution to the Pallets Loading Problem (PLP) is extremely significant. In general, there are two broad classes of PLP based on the sizes of the loaded boxes [2].

The first is the uniform PLP, which is generally applied for identical or homogeneous types of boxes, whereas the second is termed the mixed PLP, which categorizes non-identical types of boxes [3]. Furthermore, this classification of boxes can be further categorized on the basis of the box sizes in the system, termed as weakly heterogeneous and strongly heterogeneous [4,5]. Additionally, as a rule, there can be no overlap and overhang for the boxes to be loaded onto the pallets.

The Pallet Loading Problem is undeniably a complex and multifaceted task [6]. Barros et al. [6] studied the complexity of an arrangement of cargo in a pallet and proved through principal component analysis and multiple linear regression that the number of boxes is a strong predictor of complexity in this challenging problem. Moreover, the Distributor's Pallet Loading Problem (DPLP) is an NP-hard problem and has multiple layers of complexity stemming from various constraints imposed on it, such as the stability of a layer to sustain upper layers, the compression limit of each layer to sustain the weight of the layers placed on top of it, and the weight limit imposed by the company carrying out the packaging [7]. Nonetheless, loading boxes onto a container in such a manner that the 3D volume is utilized to the maximum is also a computationally demanding problem since it requires addressing various constraints, such as the orientations in which a box can be loaded as well as stability constraints [8]. Finally, cargo security during the transportation process needs to be thoroughly accounted for, and it can be enhanced through the selection of a proper stretch film and an appropriate cargo wrapping model [9].

For many industries, logistics activities such as transportation and storage play a key role in delivery, cost reduction, and the safety of goods. For this purpose, pallets are utilized as the most significant strategic equipment in the logistics of shipment and delivery. Similarly, another important factor is to utilize an optimal number of pallets for any freight, which may save an enormous amount of costs [1]. Consequently, many recent studies have focused on solving the PLP for different 3-dimensional (3D) pallet sizes to reduce the total number of pallets as well as the optimal sorting of boxes in an efficient manner [10]. Other studies have focused on the use of Artificial Intelligence (AI) and Machine Learning (ML) methods to schedule and control freight loading operations, supply chain management, and logistics in the manufacturing industry [11–14].

This research study originated as a part of the research initiative by the DHL supply chain, which focuses on determining the optimal classification methods to load boxes on the pallets, also taking into account the humidity and storage duration. It was reported that the stability, the strength of the pallet, and the mechanical strength of the box are adversely affected by changes in humidity [15]. DHL operates in more than 220 countries and territories globally, and it is one of the world's major shipping companies. It delivers goods to more than 120,000 destinations from its 5000 offices, using more than 76,000 delivery vehicles [16]. Hence, it is absolutely imperative for DHL to drive state-of-the-art research to find efficient solutions to the PLP to ensure its smooth operation. The ultimate objective of any algorithm addressing the PLP is to arrange a given number of homogeneous or heterogeneous-sized boxes in an optimized manner in terms of space or costs in a certain container or pallet. To solve a PLP, several performance measures, such as pallet utilization, the stability and strength of the pallets, the constraints related to humidity or storage duration, etc., are considered, irrespective of the aforementioned categories. Generally, the stability of the pallets increases when their utilization improves. Therefore, many studies have focused on maximizing the utilization of pallets to improve overall stability. In general, it was reported that the multi-pallet loading problem is an NP-hard problem, and researchers have used non-linear integer programming or stochastic search methods to solve it [5]. The non-linear method guarantees an optimal solution, but it is a computationally expensive method, especially if the parameters in the formulation increase [17]. Contrarily, stochastic search methods provide a sub-optimal solution at a reduced computational cost [5]. In this paper, a novel ML approach was employed for solving PLP to provide a suggestion framework to evaluate the optimal position of a box in a pallet. In the proposed methodology, two -phase algorithm is developed based on

a dataset from the DHL supply chain to load identical boxes onto the pallet. The first phase maximizes the number of boxes per horizontal layer, where five heuristics, one block, two-block, three-block, hollow-block, and G5 heuristics are utilized. The second phase determines the number of the horizontal layers loaded per pallet based on the maximum height, maximum weight and dynamic strength. More specifically, various ML methods, such as Support Vector Classifier (SVC), Decision Tree Classifier (DTC), K Nearest Neighbour Classifier (KNN), Random Forest Classifier (RFC), and Artificial Neural Network Classifier (ANN), were used on the dataset, and the categories of the existing pallet types were evaluated. This ML-based approach to solve PLP provides better results compared with running various genetic algorithms or heuristics each time. The results show that the optimal pallet selection process may take several hours when executing these traditional algorithms, compared with ~1 min after employing the ML methods. The results of this study report that a very high (>85%) Pallet Utilization Volume (PUV) can be obtained using ML models on the given dataset. Moreover, an accuracy of >89% was found to predict the accurate loading arrangement of boxes on a suitable pallet.

The remainder of the paper is organized as follows. A literature review is presented in Section 2, providing the background research studies and methods. In Section 3, the research methodology and classification methods are elaborated. Subsequently, the data analysis is carried out in Section 4. The results are presented and explained in Section 5. Finally, Section 6 provides the conclusion and future work.

## 2. Literature Review

The PLP is used to enhance performance measures, such as the number of boxes per pallet (or pallet utilization) and stability, by arranging boxes on the basis of their dimensions and weights on a rectangular pallet with known dimensions and weight limits. In the case of a 3D manufacturer's PLP for assigning the maximum number of boxes per pallet, an optimized solution depends upon the pallet stability on the number of horizontal layers per pallet and the layout pattern of the boxes per layer [10]. In the literature, there are multiple methods for solving the PLP, such as classical mathematical methods, evolutionary methods, other intelligent optimization methods, and hybrid methods. Dowsland [18] developed a pallet optimization system to place boxes on pallets, which identified an optimal solution to the PLP within ~5 min. However, the proposed solution was not efficient for larger problems. In another study, the binary integers-based formulation was used to develop a mathematical model to arrange multiple pallets and materials compatibility with those pallets [19]. In this model, the box dimensions were not necessarily integers. Similarly, another mathematical model based on a polynomial time algorithm was developed for the 2-dimensional (2D) guillotine cutting stock problems [20]. In another similar study, Ahn et al. [21] developed a mathematical model with a staircase structure that had geometrical properties. The proposed model provided reasonable efficiency in the PLP by identifying an optimal solution. Contrary to all these classical solutions to the PLP, Schuster et al. [22] optimized the stack stability in addition to the stack volume. An important feature of this work was minimizing the number of products of various customers on the same pallet. However, the proposed model can optimally solve only small-scale problems, and it becomes computationally expensive for large problems. Furthermore, Chan et al. [23] developed a two-phase intelligent Decision Support-System (DSS) for the Air-Cargo Loading Problem. It is an efficient solution for variable-sized (or shaped) pallets to be uploaded on the Air-Cargo. This particular solution has been effectively utilized in airport cargo systems, and it successfully provides over 90% volume utilization. The common factor in all these methods is that they are based on classical mathematical models. It is evident that these models should address the PLP by performing an exhaustive search of all possibilities, which become impractical for large problem sizes. Hence, the researchers looked for alternate solutions, such as genetic algorithms or heuristics, to find faster and sub-optimal solutions.

The heuristic algorithms were also utilized in the literature to solve the PLP until the mid-1990s. It was ascertained whether they can provide optimal results with the new upper-bound methods, which are based on some structural constraints [24]. For instance, Bischoff et al. [25] used a new heuristic algorithm to solve distributors' pallet packing problems efficiently and achieved the natural balance. Another distributor's pallet packing problem was solved with the heuristic methods by Terno et al. [1]. In the study, a layer-wise loading strategy with the optimal 2D loading patterns was employed in the 3D solution method. Similarly, [26] developed an effective heuristic algorithm that addressed the issue of the 2D PLP. Their novel method could readily identify complicated solutions with five-block structures.

Although heuristic algorithms are utilized extensively, there are several other intelligent optimization and search methods as well. For instance, the new branch-and-bound algorithms are integer programming-based methods, which are employed to optimize the number of boxes to be loaded on a pallet. It is an efficient method that has solved over three million problems for an area ratio bound of less than 101 boxes by executing the optimal solution in less than a minute [21]. Another study on the PLP solution was carried out by Bhattacharya et al. [27] using an exact algorithm based on the depth-first strategy. Their novel methodology was based on the idea of the reduction of the size of the search tree. Thus, the final solution effectively utilizes the available system memory. In this way, more effective pruning can be performed. In the literature, there are several hybrid methods as well, which combine the classical and heuristics methods. For instance, a hybrid approach coalescing the heuristic and genetic algorithms was developed by Lau et al. [5] to solve the PLP for profit optimization. Subsequently, this hybrid approach was compared with two stochastic search methods, Simulated Annealing (SA) and Tabu search (TS), as well as a nonlinear integer programming-based branch-and-bound method. The results of the comparative analysis revealed that the proposed hybrid method was more cost-effective than stochastic search methods. In another research study, a hybrid method was utilized in Martins and Dell's [28] study, which employed new bounds, heuristics, and exact algorithms. The proposed method was able to determine an optimal solution for approximately 99.7% of the problems. The remaining 0.3% comprised of just one box varying from the optimal solution. Li et al.'s [29] study aimed to revolutionize the warehouse design through elevating the routing optimization problem of two forklifts operating in a four-door warehouse, considered as a quadratic assignment problem. Sahin-Arslan and Ertem [30] investigated how and at what cost freight containers could be used as an inventory-holding mechanism for humanitarian logistics.

Most of the above-mentioned solutions to the PLP have the limitations of either being computationally expensive for large problem sizes or providing sub-optimal solutions. In our proposed solution for the PLP, an ML-based two-phase algorithm was developed in order to solve the 3D PLP comprising of identical boxes. In addition, the proposed solution considers the storage time and humidity without any pallet overlap and overhang [10]. Furthermore, in our proposed study, five heuristics, i.e., the one-block heuristic, three-block heuristic, five-block heuristic, hollow block heuristic, and G5-heuristic, were used to evaluate the total number of identical boxes per horizontal layer and the corresponding box loading layout or pattern on the pallet [28,31,32]. As an addition to the second phase of our previous study [6], the number of horizontal layers per pallet is computed on the basis of the dynamic compressive strength of the pallet for each of the heuristics. Consequently, the total number of boxes per horizontal layer and the total number of horizontal layers are used to determine the number of boxes per pallet for each of the aforementioned five heuristics. In the final stage, the box-loading layout that furnishes the maximum number of boxes per pallet is preferred. To the best of our knowledge, no prior study evaluated the solution for the PLP using ML methods. The models were trained and tested on real-world data acquired from the DHL supply chain; therefore, the results of this study can be significantly utilized in any similar research study. We could not compare the results since

no study using traditional optimization could be found in the literature for pallet loading. Therefore, we believe that this is an original contribution to the literature.

## 3. Methodology

In this section, the methodology utilized for the proposed research is described. Figure 1 shows the methodology of this study. It should be noted that the relevant methods and performance measures given in the literature are significant for evaluating the methods utilized to solve the PLP in this paper. In addition, such a comprehensive evaluation provides a basis for the comparison of various methods. Our methodology is described in the following subsections.

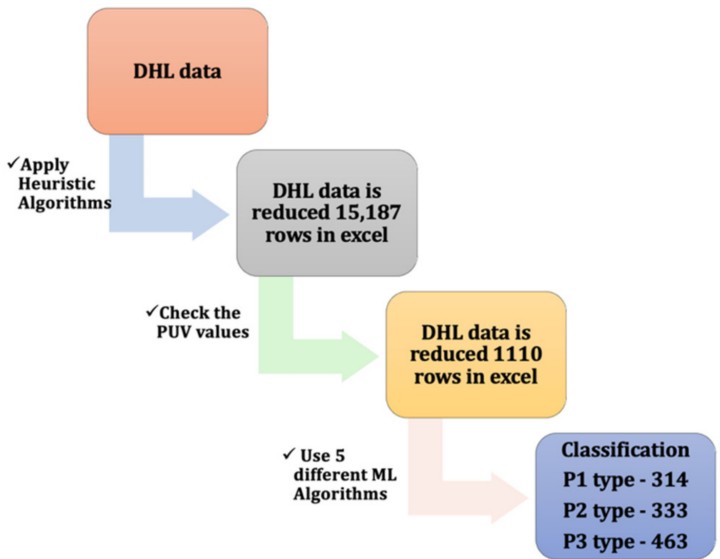

**Figure 1.** Flow chart of the proposed approach.

### 3.1. Number of the Boxes Loaded onto the Pallet

The process to ascertain the number of identical boxes to be placed on a typical pallet was divided into two phases. The dimensions of a typical box are coded as a tuple $(l, w, h,$ and $m)$, where $l$ is the length of the box, $w$ is the width of the box, $h$ is the height of the box, and $m$ is the weight of the box. Similarly, the dimensions of the pallet are defined as a tuple $(L, W, H_{max},$ and $M_{max})$, where $L$ is the pallet length, $W$ is the pallet width, $H_{max}$ is the maximum allowable pallet load height, and $M_{max}$ is the maximum allowable weight on the pallet, in such a way that $(L \geq l, W \geq w)$. In the first phase, the total number of boxes loaded on the horizontal layer is identified. These boxes should be placed completely onto the pallet without any overlap or overhang. If the value of (pallet area)/(box area) is less than 101, many studies [25,28,31–33] have shown that various heuristics, such as one-block, three-block, five-block, hollow-block, and G5-heuristics, can determine the maximum number of boxes to be arranged in each horizontal layer. In the second phase, the total number of horizontal layers per pallet is calculated on the basis of $H_{max}$. Both these phases are elaborated as follows.

### 3.1.1. Phase I: Determining the Box Arrangement on a Horizontal Layer

In this phase, different heuristics, such as one-block, three-block, five-block, hollow-block, and G5-heuristics, are implemented to arrange the boxes on a horizontal layer with known and fixed dimensions. Any of these five heuristics can yield the maximum number of boxes in the layer. Furthermore, three partition methods, termed *feasible* partition, *efficient* partition, and *perfect* partition, are utilized to fit the boxes completely on the pallet without any overlap or overhang. These methods are explained as follows:

Assuming that $(q, n)$ denotes an ordered pair of non-negative integers satisfying the condition $q * l + n * w \leq S$ for a pallet dimension $S$ ($L$ or $W$), then the ordered pair $(q, n)$ is known as a *feasible* partition of $S$. If $q$ and $n$ also satisfy the condition $0 \leq S - q * l - n * w < l$, then $(q, n)$ is called an *efficient* partition of $S$. Similarly, if $q$ and $n$ satisfy the condition $q * l + n * w = S$, then $(q, n)$ is called a *perfect* partition of $S$. The set of all *feasible* partitions of the pallet dimension $S$ is denoted as $F(S, l, w)$. The set of all *efficient* partitions of the pallet dimension $S$ is denoted as $PE(S, a, b)$, and the set of all *perfect* partitions of the pallet dimension $S$ is denoted as $P(S, l, w)$ [34,35].

In the one-block heuristic, the boxes are loaded in either H-box or V-box orientation on the basis of the ratio between the dimensions of the pallet and boxes without any overhang. For example, identical boxes with known dimensions are loaded onto the pallet; $h$ is a dimension perpendicular to the surface of the pallet, as depicted in Figure 2. If $\left[\frac{L}{l}\right] * \left[\frac{W}{w}\right] > \left[\frac{W}{l}\right] * \left[\frac{L}{w}\right]$, where the length of the boxes is parallel to the length of the pallet, then the horizontal layout pattern for the boxes in this layer is known as the H-box pattern; otherwise, if the length of the box is parallel to the width of the pallet, the box vertical layout pattern is known as the V-box pattern. All boxes in the layer should be loaded into the layer in the uniform orientation, which means that all boxes must have either an H-box or V-box layout. There are also three possibilities to load the boxes in the layer, where either $l, w \vee h$ is perpendicular to the surface of the pallet.

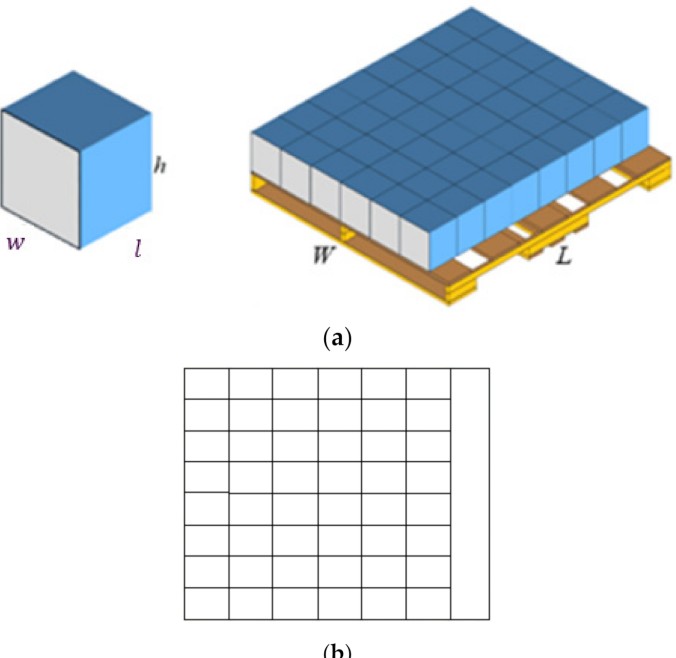

(a)

(b)

**Figure 2.** One-block heuristic. (**a**) Arrangement of boxes per horizontal layer using the one-block heuristic. (**b**) Top view for the horizontal layer using the one-block heuristic.

In the three-block and five-block heuristics, the layout of the layer is divided into three and five blocks, respectively (Figure 3). The boxes in each block should be arranged uniformly in either an H-box or V-box orientation [31,36].

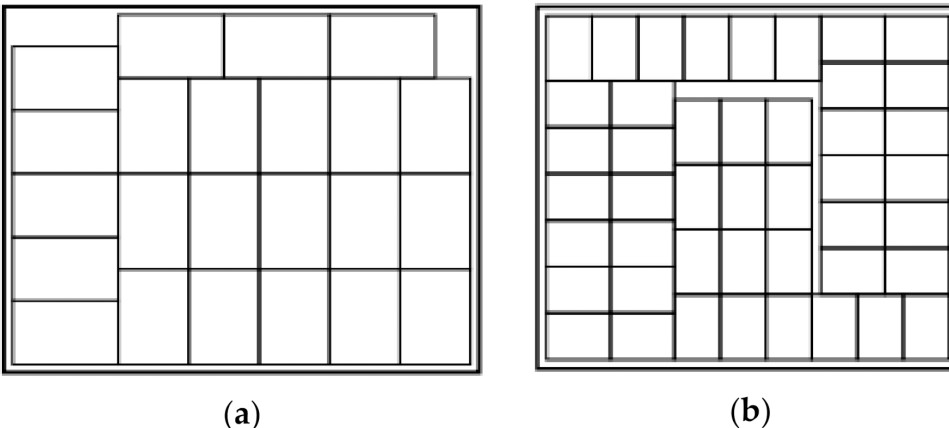

(a)           (b)

**Figure 3.** The three-block and five-block heuristics: (**a**) three-block heuristic and (**b**) five-block heuristic [36].

The hollow block heuristic can also be utilized to load identical boxes on the horizontal layer. The heuristic divides the layer layout into diagonal and main blocks, as presented in Figure 4. Diagonal blocks ($DB$) and main blocks ($MB$) are created with boxes loaded in different directions, where these blocks should cover the entire length and width of the pallet [28]. The boxes' orientation in each block is identical.

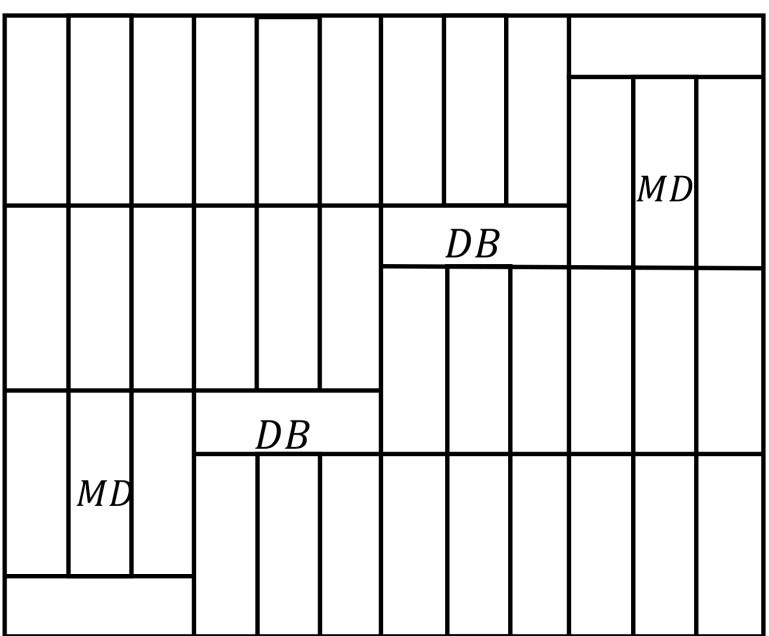

**Figure 4.** Hollow block [28].

The G5-heuristic is one of the heuristics in the PLP literature frequently used in order to maximize the total number of boxes loaded into a horizontal layer and minimize the waste area. In this heuristic, the layout of the layer is divided into five different blocks (four blocks in the corners and one in the center). The dimensions of the central block decrease if the dimensions of the other four blocks in the corners increase, and vice versa. Each block is loaded independently, and the loading process can be performed by any of the other four heuristics, namely the one-block, three-block, hollow block, and five-block heuristics. The total number of boxes per layer is calculated on the basis of the sum of the maximum number of boxes generated for each block [28]. Once the maximum number of boxes per

horizontal layer is calculated using the five aforementioned heuristics, the optimal number of boxes in the layer is selected for three different possibilities based on h, b, and a as the perpendicular dimension to the pallet in each case.

### 3.1.2. Phase II: Computing the Number of Horizontal Layers on the Basis of the $H_{max}$

In the second phase, the total number of boxes per pallet and the number of the horizontal layers per pallet are determined considering three parameters: (a) the total height of the horizontal layers should not exceed the $H_{max}$; (b) the total weight on the pallet should not be more than $M_{max}$; and (c) the average load on the box at the bottom horizontal layer of the pallet should be less than its dynamic compression strength, considering humidity, interlock stacking pattern, dimensions of the box, and storage time. The $H_{max}$ is determining either on the basis of storage on pallet within the warehouse, i.e., racks, floor, etc., or on the basis of transportation constraints, i.e., trailer internal height, number of horizontal layers loaded, etc. In this case, the height of the total number of the horizontal layers should be less than the $H_{max}$ since more than one horizontal layer can be used per pallet (Equation (1)). This paper assumes that all the layers have the same pattern, where overlap between layers is not allowed; therefore, all the horizontal layers are identical and have the same height, as depicted in Figure 5.

$$NHL_{max} = \left\lfloor \frac{H_{max}}{BDPB} \right\rfloor \tag{1}$$

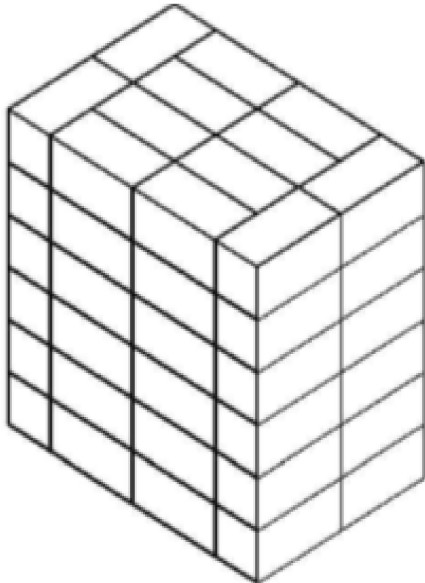

**Figure 5.** Pallet Loading Problem with identical horizontal layers based on $H_{max}$.

*NHL*: The maximum number of horizontal layers
*BDPB*: Box's dimension perpendicular to the base

Furthermore, the total weight on the pallet should not be more than $M_{max}$. This is important for several reasons. First, if there is no height limit, the number of boxes per pallet will be determined based on the $M_{max}$ per pallet. Second, the total weight of the boxes per pallet when loading on the shelves or the trailers should be less than the maximum weight capacity to avoid damage to the pallets, the shelves, and the trailers.

Compression strength is another parameter that is used to define the number of horizontal layers per pallet to maximize the total number of boxes loaded. Compression strength represents the capacity of a material or structure to withstand loads without any deterioration. For a pallet, compression strength is the maximum weight that the boxes in

the bottom layer can support without any damage. In this section, pallet dynamic compression strength is calculated by modifying the McKee formula by adding considerations for humidity, storage time, interlock stacking pattern, and the pallet shape coefficients. The McKee formula is usually used to determine the static compression strength, which is a theoretical value under ideal laboratory conditions, with the temperature and humidity controlled within $73 \pm 2\,°F$ and $50\% \pm 2\%$ RH, respectively [36]. On the other hand, under real conditions, dynamic compression strength is the actual compression strength of a box. Equations (2) and (3) show the static compression strength and dynamic compression strength. $ECT$ is the Carton Edge Crush Test, which measures the ability of a corrugated board to sustain a top-to-bottom load [37]. $CAL$ is caliper of the corrugated board, $PER$ is box perimeter from the top view, $S_S$ is the static compression strength, $F_O$ is the orientation coefficient form strength, $F_T$ is the storage time coefficient, $F_I$ is the interlock coefficient, $F_H$ is the humidity coefficient, and $F_G$ is the pallet shape coefficient (Table 1)

$$\text{Static Compression Strength } (S_S) = 5.874 * ECT * CAL^{0.508} * PER^{0.492} \qquad (2)$$

$$\text{Dynamic Compression Strength } (S_D) = S_S * F_T * F_H * F_G * F_o * F_I \qquad (3)$$

**Table 1.** Dynamic strength factors [38].

| Storage Time ($F_T$) | | Humidity ($F_H$) | | Pallet Surface Gapped ($F_G$) | | Interlock ($F_I$) | | Perpendicular Dimension to Base ($F_o$) | |
|---|---|---|---|---|---|---|---|---|---|
| 0 | 1 | 0–0.45 | 1.1 | Yes | 0.92 | Yes | 0.60 | 1st Shortest Dimension of Box | 1 |
| 1–3 | 0.7 | 0.45–0.55 | 1 | No | 1 | No | 1 | 2nd Shortest Dimension of Box | 0.9 |
| 4–10 | 0.65 | 0.55–0.65 | 0.9 | | | | | Longest Dimension of Box | 0.8 |
| 11–30 | 0.6 | 0.65–0.75 | 0.8 | | | | | | |
| 31–90 | 0.55 | 0.75–0.85 | 0.7 | | | | | | |
| 91–120 | 0.5 | 0.85–1 | 0.5 | | | | | | |
| 121–300 | 0.45 | | | | | | | | |

### 3.2. Machine Learning Methods for Classification

Classification is a technique employed in AI and ML that is used to label a given dataset into different classes in order to extract useful information [39]. Currently, classification methods are used in many fields, such as speech recognition [40], image classification [41], and document classification [42]. Fundamentally, there are two classification methods, which are binary and multi-class classifications [43]. In the binary classification, a given problem is divided into two situations or classes, such as male/female in gender prediction [44] and sick/not sick in disease prediction [45]. On the other hand, multi-class classification is used in problems in which there are more than two cases, such as estimating different flower types [46]. There are classification methods such as ANNs, Support Vector Machines (SVMs), KNN, Naive Bayes (NB), Decision Trees (DT), and Random Forest. In this study, all these classification methods are used because of their success and frequent use in recent research studies.

In the process of the development of ANNs, researchers were inspired by the human nervous system [47]. These ANNs consist of input, output, and hidden layers, and they are trained and tested on a real dataset to classify the data into different classes. Later on, the trained ANN models try to predict the outcome of any new data for further analysis [48]. This whole process is governed by the neurons in the layers and an activation function. This activation function (also called the objective function) is a predefined goal for a particular ANN model.

The SVM enables the use of vectors (separators) to divide the training data into areas (categories) as far away as possible [49]. It assigns the test data a certain category according to the area of a particular vector that they fall into and uses a subset of training points when making decisions in the classification process. For this reason, the SVM process uses memory efficiently and is effective for high-dimensional data.

The KNN method classifies an object in the input parameter field with the majority vote of the object's neighbors [50]. In this way, an object is assigned to the most common class among its nearest neighbor, i.e., k (user-specified integer). This method classifies objects according to their similarity in the point of property. It is a simple method to implement and provides effective results in large test datasets [51].

The DT method is a flowchart-like tree structure in which an inner node represents the property (or attribute), a branch represents a decision rule, and each leaf node represents the result [52]. The top node in a DT is termed the root node. It learns to partition according to the attribute value and sections the tree recursively with the recursive partitioning call. This flowchart-like structure in a DT helps in decision-making [53]. Therefore, DTs are easy to understand and interpret.

The Random Forest method is a supervised learning algorithm. It can be used for both classification and regression. It is also the most flexible and easy-to-use algorithm [54]. Similar to a forest consisting of numerous trees, the random forests generate decision trees on a randomly-selected data samples, each making a guess from the tree and choosing the optimal solution by voting [55]. Furthermore, this method provides a reasonably good indication of the feature's importance.

### 3.3. Evaluation Metrics

The evaluation metrics are typically used to quantify, analyze, and test the success of classification and estimation methods [56]. Global metrics such as overall accuracy, overall balanced accuracy, Cohen's Kappa, Matthew's Correlation Coefficient extended to multi-class, and confusion entropy ensure the quantification of multi-class classification and enable the comparison of various classification methods and techniques [57].

In the classification process, the ability of a model to perform a correct classification is quantified with the accuracy value [58], which is directly computed from the confusion matrix [57].

$$\text{Accuracy} = \frac{TP + TN}{TP + TN + FP + FN} \tag{4}$$

Accuracy is a very intuitive metric that enables an overall understanding of the performance of a model on the test data [57]. However, when dealing with imbalanced datasets, accuracy has a tendency to conceal strong classification errors for classes with a limited number of units [57]. In this case, accuracy does not enable the identification of classes for which the model has suboptimal performance [57].

In the case of imbalanced datasets, balanced accuracy ensures that the prediction accuracy is not inflated because one or several classes have results that dominate the other classes [59]. This measure represents the arithmetic mean of the performance of a model for each class [59]. It is a balanced metric in the sense that it assigns the same weight and thus the same importance to each class studied [59].

$$\text{Balanced Accuracy} = \frac{1}{C} \times \sum \frac{p_i}{n_i}, \tag{5}$$

where $p_i$ is the correctly predicted number of data items in class $i$ and $n_i$ is the true number of data items in class $i$.

As can be inferred from the formula below, the Matthews Correlation Coefficient uses all the entries of the confusion matrix both in the numerator and in the denominator. Because of this fact, it can generally be considered a balanced indicator of performance [59]. This measure takes values in the range [–1, 1], with values close to 1 indicating a strong prediction and correlation between the predicted and true labels. Values close to $-1$ indicate an inverse relationship between the predicted and true labels of the dataset and reflect systematic errors in the modeling process. In the case of values close to 0, the classifier makes random decisions when assigning the labels to the classes [59].

Matthews Correlation Coefficient for multi-class classification:

Given multi-class Confusion Matrix C:

$$K = \frac{c \times s - \sum p_k \times t_k}{\sqrt{\left(s^2 - \sum p_k^2\right) \times \left(s^2 - \sum t_k^2\right)}} \tag{6}$$

$c = \sum C_{kk}$ the total number of elements correctly predicted; $s = \sum \sum C_{ij}$ the total number of elements; $p_k = \sum C_{ki}$ the total number of times class $k$ was predicted; $t_k = \sum C_{ik}$ the total number of times that class $k$ actually occurs.

Cohen's Kappa for multi-class cases:

The Cohen's Kappa Score possesses similarities to the Matthews Correlation Coefficient in the case of multi-class classification and leverages the expected accuracy, a measure that reflects the dependence obtained by chance between the predicted and the true labels [59]. The coefficient quantifies the agreement between the predicted and actual classes as follows: when $K = 0$, the model's prediction is independent of the actual classification, when $K = 1$, the model's prediction depends entirely on the actual classification, and when $K = -1$, there is no agreement between the model's prediction and the actual classification [59].

Given multi-class Confusion Matrix C:

$$K = \frac{c \times s - \sum p_k \times t_k}{s^2 - \sum p_k \times t_k} \tag{7}$$

$c = \sum C_{kk}$ the total number of elements that were correctly predicted; $s = \sum \sum C_{ij}$ the total number of elements; $p_k = \sum C_{ki}$ the total number of times class $k$ was predicted; $t_k = \sum C_{ik}$ the total number of times that class $k$ actually occurs.

The cross entropy quantifies the difference between two probability distributions, and it can be calculated using the formula below [60,61].

$$\text{Cross Entropy} = -\sum Likelihood_{Reference}(i) \times log_2 Likelihood_{Response}(i) \tag{8}$$

Sensitivity by class, specificity by class, precision by class, F1 by class, balanced accuracy by class, and confusion entropy by class enable us to understand the performance of different classifiers with respect to the classes being analyzed [60].

Sensitivity, also called the true positive rate, captures the proportion of positive outcomes that are correctly identified as such [60,62]. *TP* represents the prediction made when the model is able to correctly predict a positive outcome assigned to the correct class, whereas an *FN* occurs when a false prediction is assigned to the correct class [63].

$$\text{Sensitivity} = \frac{TP}{TP + FN} \tag{9}$$

Specificity, also called true negative rate, captures the proportion of negative outcomes that are correctly identified as such [60,62]. In the case of *TN*, the false prediction is assigned to the incorrect class, while *FP* represents the correct estimate assigned to the incorrect class [63].

$$\text{Specificity} = \frac{TN}{TN + FP} \tag{10}$$

Precision, also called positive predicted value, represents the proportion of positives that corresponds to the presence of a certain condition [60,62].

$$\text{Precision} = \frac{TP}{TP + FP} \tag{11}$$

The F1-macro score is the harmonic mean of sensitivity and precision and is calculated according to the formula below [57,60]. In the computation of the F1-macro score, the largest classes are assigned the same importance as small classes [57], which makes the

F1-macro score a suitable indicator of the performance of a machine learning algorithm on an imbalanced dataset. A high F1-macro value means that the algorithm performs well on all classes under consideration, whereas a low F1-macro score is an indicator of poor prediction of the classes analyzed [57].

$$\text{F1} - \text{macro} = \frac{2}{|C|} \sum \frac{\text{Sensitivity}_i \times \text{Precision}_i}{\text{Sensitivity}_i + \text{Precision}_i} \tag{12}$$

The balanced accuracy represents, in essence, an average of recalls. It is computed according to the formula below [57].

$$\text{Balanced Accuracy} = \frac{\text{Sensitivity} + \text{Specificity}}{2} = \frac{\left( \frac{TP}{TP+FN} + \frac{TN}{TN+FP} \right)}{2} \tag{13}$$

The confusion entropy indicates how well samples belonging to different classes have been separated from each other. It exploits the misclassification of information of confusion matrices in order to quantify the confusion level of the class distribution of misclassified samples [60].

$$CEN_j = - \sum \left( P_{j,k}^{|C|} log_{2(|C|-1)} \left( P_{j,k}^{j} \right) + P_{k,j}^{j} log_{2(|C|-1)} \left( P_{k,j}^{j} \right) \right), \text{ where} \tag{14}$$

$$P_{i,j}^{i} = \frac{C_{ij}}{\sum (C_{ik} + C_{ki})}$$

where $C_{ij}$ is the entry found at position $(i,j)$ in the confusion matrix.

## 4. Application and Dataset Description

This section presents our methodology to implement the ML models to solve the PLP. The ML models were trained and tested on a real dataset. Moreover, a comprehensive evaluation strategy was employed in order to compare the results with various ML algorithms.

### 4.1. Dataset

In this paper, the dataset comprised 15,187 data points, which contained various types of boxes and pallet-sizes. This dataset was used for the training and testing of different models, which utilized the aforementioned five heuristic algorithms to place the boxes on pallets according to their variable lengths, heights, and widths to obtain optimal results for their arrangements. This dataset is available as listed within an Excel file. The pre-processing of the dataset was first performed by data filtering to create a meaningful classification and eliminate unnecessary data points. The pre-processed dataset is extremely important for the proper training and testing process of the ML models. For instance, from this dataset, the data in accordance with the most commonly-used standard pallet sizes 42 × 42 (P1), 45 × 45 (P2), and 48 × 40 (P3) were selected for training and testing. Thus, our algorithm is applicable only for the P1, P2, and P3 pallet sizes, with only approximately 1 in 15 instances being further considered. While selecting these data, the PUV (pallet utilization volume) ratio for both the P1 and P2 pallet sizes was 85%, and the PUV ratio for the P3 pallet was 95% or above. Thus, a total of 1110 data points were finalized, with 314 for the P1 pallets, 333 for the P2 pallets, and 463 for the P3 pallets. Table 2 presents the dataset details used in this research study.

**Table 2.** The datasets used in model training and testing.

| Total Data Points | Data Points after Pre-Processing | P1 Pallets | | P2 Pallets | | P3 Pallets | |
|---|---|---|---|---|---|---|---|
| | | Training | Testing | Training | Testing | Training | Testing |
| 15,187 | 1110 | 283 | 31 | 299 | 34 | 417 | 46 |

*4.2. Implementation Details*

All the ML models were trained and tested on a uniform dataset; 10-fold cross-validation was used for the 1110 data points for which the algorithm can be applied. Since we were dealing with an imbalanced data set, we first split the data into 10 folds, with 9 folds used for training and the remaining fold used for testing. Since we were dealing with a limited amount of data, we used an oversampling technique called Synthetic Minority Over-Sampling Technique (SMOTE) on the testing dataset in order to address the imbalanced class problem by oversampling the under-represented classes [64]. We used the SMOTE technique implemented in the Python imblearn package [65]. We computed the required metrics using the Python packages scikit-learn [66] and PyCM [60]. For each metric, the average result over 10 rounds of cross-validation is reported. The metrics were computed according to Equations (4)–(14).

All classification models were implemented in Python 3.8 and were executed for training and testing on the given dataset on a computer with Intel i7 2.4 GHz CPU and 8 GB RAM.

The classification models used in this study required a number of parameters for the tuning before the actual training phase was initiated. These parameters for each classifier are described in detail as follows. The values of these standard parameters were optimized in order to acquire better results during the training and testing phases.

For the ANN classifier, since there are three target classes (P1, P2 and P3 pallets) and four inputs (box width, length, height value and demand), a network structure with four inputs and three outputs was created, and two hidden layers each containing 10 neurons were utilized. For the activation function, sigmoid was used. Moreover, the "adam" solver was chosen, with max_iter = 10,000. The random state was set to 0. For the DTC, the parameters max depth = none, min samples split = 2, random state = none, and min_samples_leaf = 1 were selected. For the KNN classifier, the parameters n_neighbors = 5, weights = "uniform", and algorithm = "auto" were chosen. For the Random Forest classifier, the parameters n_estimators = 100, max_depth = none, min_samples_leaf = 1, and random_state = none were used. For the SVM classifier, the parameters kernel = "rbf", degree = 3, gamma = "scale", coef0 = 0.0, cache_size = 200, class_weight = none, verbose = false, max_iter = −1, and random_state = none were chosen.

## 5. Results and Discussion

In our previous study [10], 15 real-life datasets were obtained from the DHL supply chain, and the proposed algorithm was applied to this dataset. After a detailed analysis, an algorithm was described and used for the results of other algorithms for comparison. Thus, the results comprising the total number of boxes per pallet were compared. Consequently, an improvement of 6.7% was reported compared with the previous methods by uniformly considering the same 15 real-life datasets. It was concluded that the proposed algorithm manifested an improvement over the previous state-of-the-art research.

In this research study, a current, more extensive dataset (given in Table 2) was used for training and testing with five different classification methods: ANN, DT, KNN, Random Forest, and SVM. The distribution of data according to three classes is depicted in Figure 6. It can be observed in this figure that all three classes are clustered around the box widths (*Y*-axis) and lengths (*X*-axis) of 5–20 units and 5–25 units, respectively.

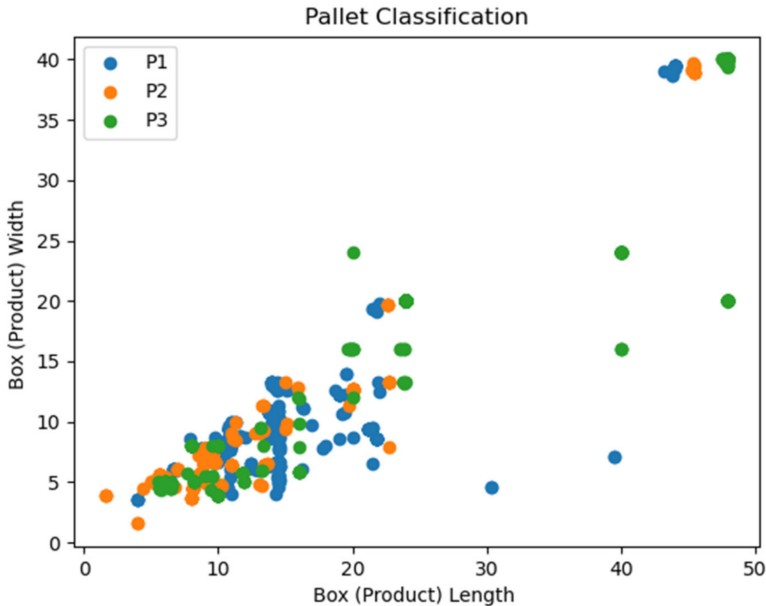

**Figure 6.** Distribution of the 1K data points according to class.

Table 3 shows that the Random Forest Classifier achieved the highest accuracy on the test dataset, namely 89%, and the highest overall balanced accuracy, namely 88%. The Random Forest Classifier was closely followed by the Decision Tree Classifier, with an overall accuracy of 88% and an overall balanced accuracy of 87%. The Support Vector Machine Classifier achieved the lowest overall accuracy (74%) and the lowest overall balanced accuracy (74%).

**Table 3.** Evaluation of the ANN, Decision Tree, KNN, Random Forest, and Support Vector Machine on the test dataset in terms of global metrics.

| Method | Accuracy | Overall Balanced Accuracy | Cohen's Kappa | Matthew's Correlation Coefficient | Cross Entropy |
|---|---|---|---|---|---|
| ANN | 0.76 | 0.76 | 0.64 | 0.66 | 1.66 |
| Decision Tree | 0.88 | 0.87 | 0.82 | 0.83 | 1.61 |
| KNN | 0.82 | 0.81 | 0.73 | 0.74 | 1.65 |
| Random Forest | 0.89 | 0.88 | 0.84 | 0.84 | 1.61 |
| Support Vector Machine | 0.74 | 0.74 | 0.62 | 0.63 | 1.67 |

Moreover, the Random Forest Classifier also ranked first in terms of Cohen's Kappa, with K = 0.84, meaning that the model's prediction depends entirely on the actual classification. The Decision Tree Classifier ranked second in terms of the Cohen's Kappa metric, with K = 0.82. The Support Vector Machine Classifier also ranked last in terms of Cohen's Kappa, with K = 0.62.

In terms of Matthews Correlation Coefficient, the Random Forest Classifier was the best performing model, showing that this classifier has high predictive power. The Decision Tree classifier ranked second (0.83), followed by the KNN (0.74), the ANN (0.66), and the SVM (0.63).

Regarding cross entropy, all the classifiers achieved similar results, with the Support Vector Machine classifier showing a slightly higher cross-entropy of 1.67.

In terms of sensitivity by class, the Random Forest Classifier achieved the best outcomes, reaching a sensitivity of 0.90 for class P1, 0.81 for class P2, and 0.94 for class P3. The Decision Tree classifier achieved comparable results, with a sensitivity of 0.90 for class P1, 0.8 for class P2, and 0.92 for class P3. The results are presented in Table 4.

**Table 4.** Evaluation of the ANN, Decision Tree, KNN, Random Forest, and Support Vector Machine on the test dataset in terms of metrics by class.

| Method | Class (Pallet) | Sensitivity by Class | Specificity by Class | Precision by Class | F1-Macro by Class | Balanced Accuracy by Class | Confusion Entropy by Class |
|---|---|---|---|---|---|---|---|
| ANN | P1 | 0.76 | 0.89 | 0.79 | 0.75 | 0.82 | 0.36 |
| | P2 | 0.72 | 0.82 | 0.65 | 0.65 | 0.77 | 0.42 |
| | P3 | 0.79 | 0.94 | 0.91 | 0.84 | 0.86 | 0.28 |
| Decision Tree | P1 | 0.90 | 0.90 | 0.84 | 0.85 | 0.90 | 0.24 |
| | P2 | 0.80 | 0.94 | 0.85 | 0.81 | 0.87 | 0.25 |
| | P3 | 0.92 | 0.96 | 0.95 | 0.93 | 0.94 | 0.15 |
| KNN | P1 | 0.79 | 0.88 | 0.78 | 0.77 | 0.84 | 0.36 |
| | P2 | 0.81 | 0.88 | 0.76 | 0.76 | 0.85 | 0.32 |
| | P3 | 0.84 | 0.96 | 0.94 | 0.87 | 0.90 | 0.20 |
| Random Forest | P1 | 0.90 | 0.91 | 0.84 | 0.86 | 0.90 | 0.23 |
| | P2 | 0.81 | 0.96 | 0.88 | 0.82 | 0.88 | 0.22 |
| | P3 | 0.94 | 0.97 | 0.96 | 0.95 | 0.95 | 0.12 |
| Support Vector Machine | P1 | 0.73 | 0.86 | 0.72 | 0.70 | 0.79 | 0.44 |
| | P2 | 0.71 | 0.85 | 0.69 | 0.64 | 0.78 | 0.39 |
| | P3 | 0.77 | 0.90 | 0.85 | 0.80 | 0.84 | 0.32 |

Additionally, the Random Forest Classifier achieved the highest specificity for all three classes considered, namely 0.91 for class P1, 0.96 for class P2, and 0.97 for class P3.

It is worth mentioning that all classifiers achieved the highest specificity by class for class P3.

In terms of precision by class, the Random Forest Classifier ranked first, with class P1 having a precision of 0.84, class P2 having a precision of 0.88, and class P3 having a precision of 0.96. The best precision by class was achieved for class P3.

The Random Forest performed best in terms of the F1-macro score as well, achieving an F1-macro score of 0.86 for class P1, 0.82 for class P2, and 0.95 for class P3.

This classifier also performed best in terms of balanced accuracy by class, with a balanced accuracy for class P1 equal to 0.90, a balanced accuracy for class P2 equal to 0.88, and a balanced accuracy for class P3 equal to 0.95, showing a high predictive power.

The Random Forest Classifier displayed the lowest confusion entropy by class, namely 0.23 for class P1, 0.22 for class P2, and 0.12 for class P3.

The Confusion Matrix graphics of all five classifiers are depicted in Figures 7–11.

In Figure 7, the Confusion Matrix for the ANN classifier is presented. It can be observed that the ANN classifier correctly classified 83% of the P1 classes, 79% of the P2 classes, and 81% of the P3 classes.

Judging from Figure 8, which depicts the Confusion Matrix for the Decision Tree, the classifier managed to correctly predict 100% of classes P1, P2, and P3.

Figure 9 shows the Confusion Matrix for the KNN classifier, which managed to correctly capture 97% of the P1 classes, 92% of the P2 classes, and 94% of the P3 classes.

In Figure 10, the Confusion Matrix for the Random Forest classifier is shown. It can be observed that this classifier managed to predict 100% of classes P1, P2, and P3.

Judging from Figure 11, which illustrates the Confusion Matrix for the SVM, the classifier managed to correctly capture 73% of the P1 classes, 75% of the P2 classes, and 77% of the P3 classes.

According to Figure 12, average PUV values in P1, P2, and P3 were 90.31, 78.67, and 82.97, respectively.

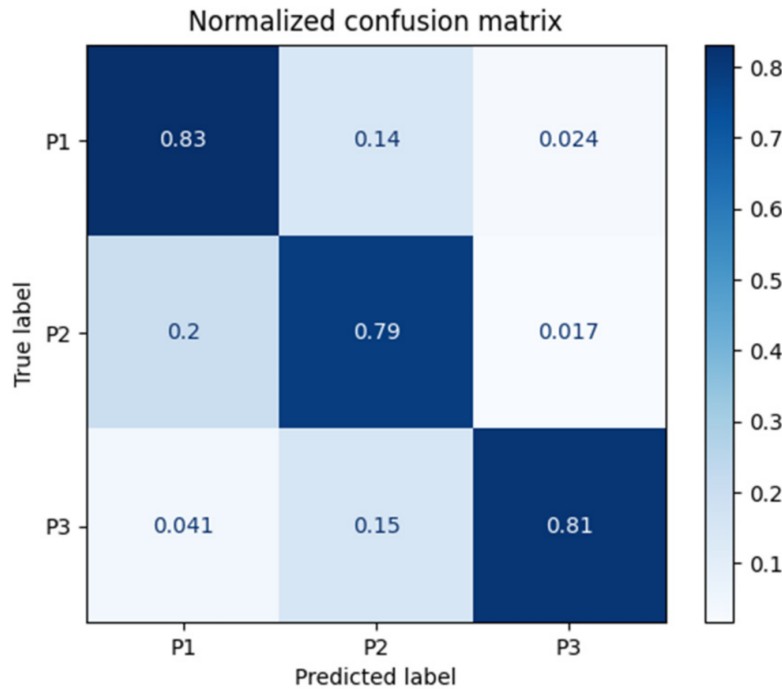

**Figure 7.** Normalized confusion matrix for ANN.

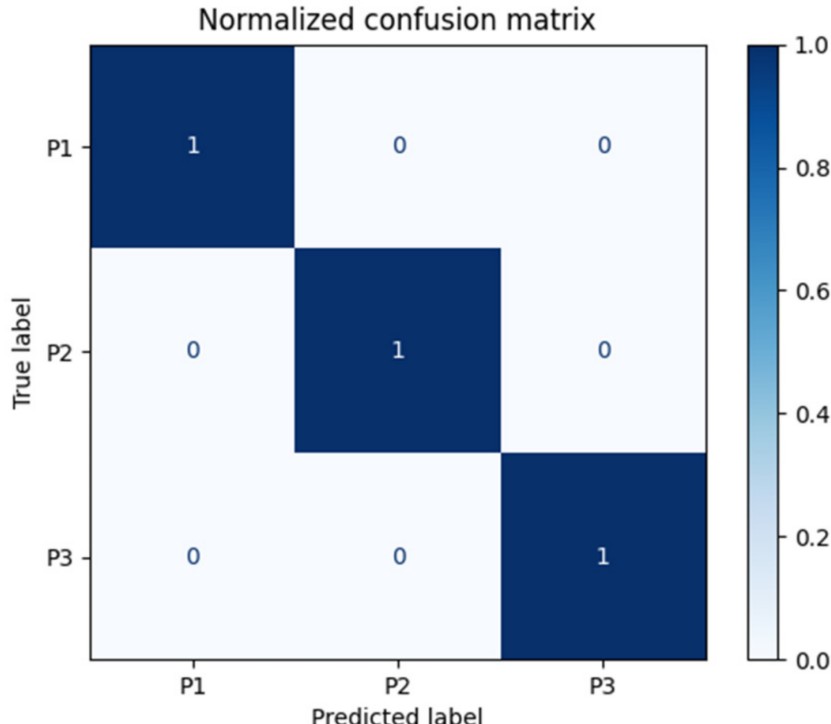

**Figure 8.** Normalized confusion matrix for Decision Tree.

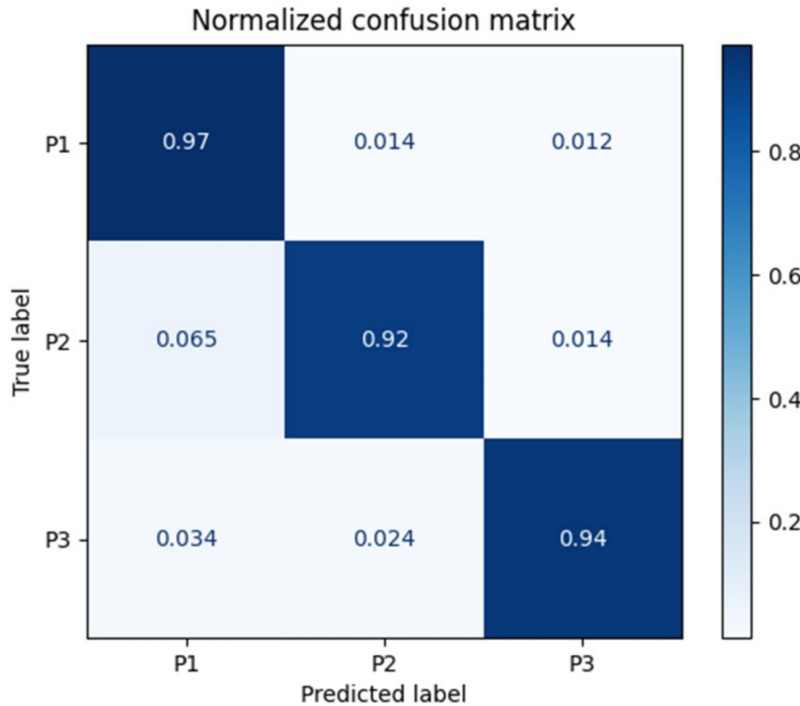

**Figure 9.** Normalized confusion matrix for KNN.

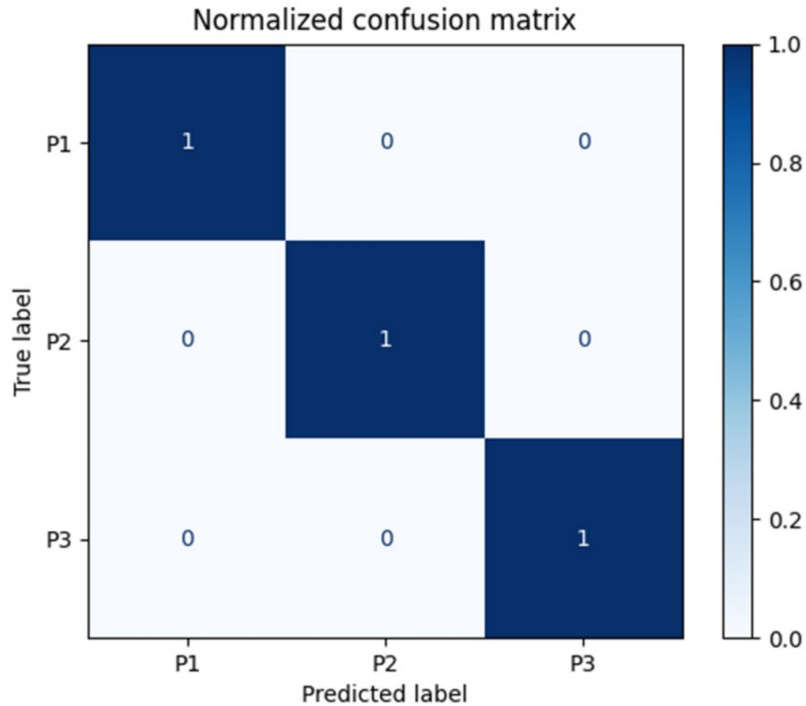

**Figure 10.** Normalized confusion matrix for Random Forest.

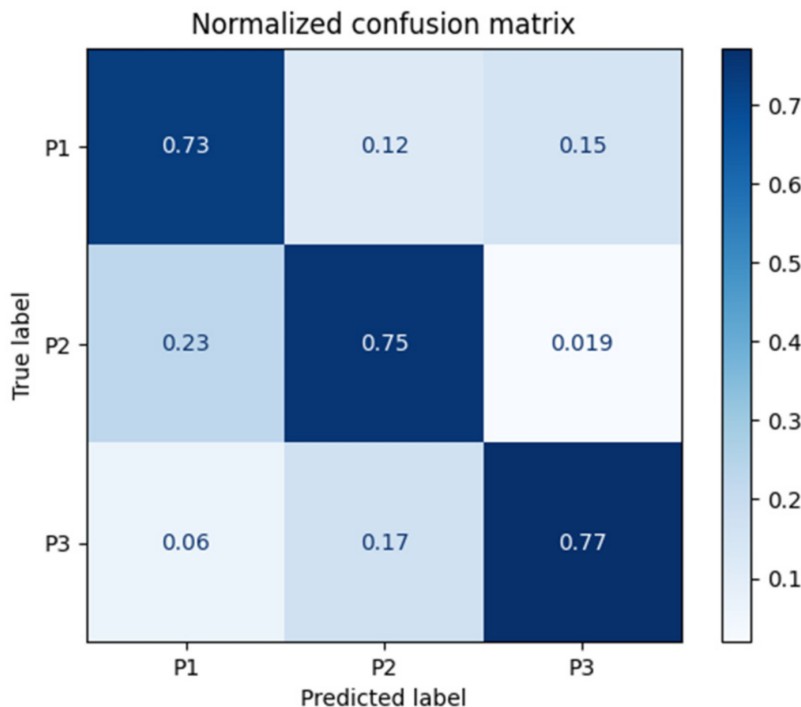

**Figure 11.** Confusion matrix for SVM classifier.

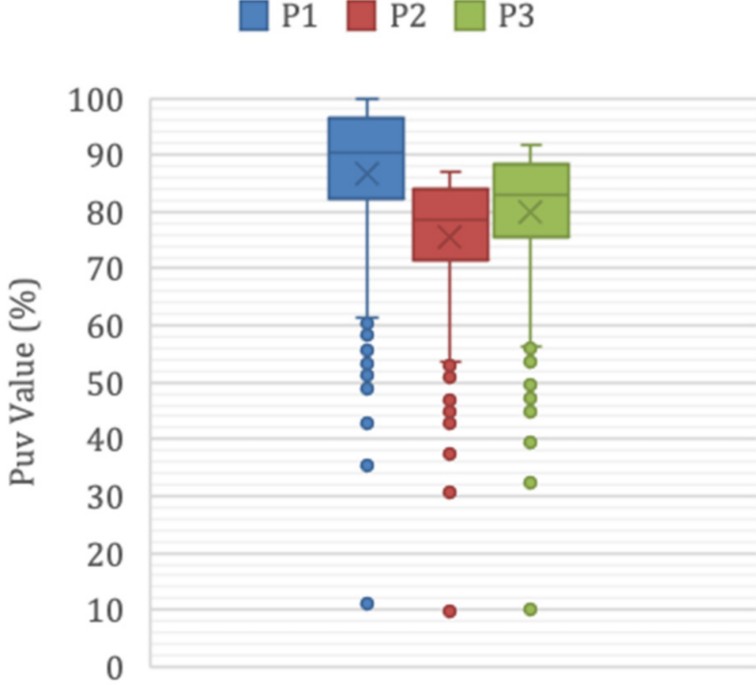

**Figure 12.** Distribution of average PUV values for the 1 K instances for which the algorithm can be applied.

The PUV values of 777 boxes were calculated for each pallet size, and the results are presented in Figure 13. It can be noted that the maximum PUV values were obtained in the case of P1 because of the smallest volume of the pallet in P1. Pallet size P3 provided the second best PUV value results because of its smaller volume compared with P1. Finally, the PUV values of the largest pallet volume (P3) were the highest because they had the largest volume.

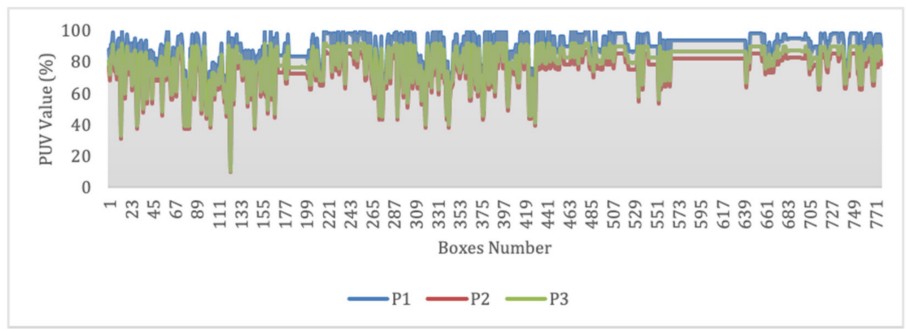

**Figure 13.** A comparison of PUV values according to pallet sizes for the 1 K instances for which the algorithm can be applied.

Overall, the variety of results given in Section 5 provide an overall idea for shipment designers to choose better pallet or box sizes for the pallet loading. Furthermore, it was observed that all the ML classifiers have high accuracy, and the designer can select an appropriate set of pallets, box sizes, and configurations using these results.

## 6. Conclusions

This paper shows an application of ML methods in a warehouse for the PLP. Because of the global competition faced by large companies such as DHL, shipment issues must be addressed in an optimal manner to save costs, time, and energy. In logistics, storage and transportation are two key elements, as they increase the company's expenditure. Reducing the storage and transportation costs may increase the profitability of the company. Subsequently, the use of pallets plays an important role in the transportation and storage of many products, and this process facilitates the efficient shipment of any product(s) by allowing them to be stored in any given facility. To solve the PLP for better facilitation, various studies have been carried out over time. In these studies, different methods have been attempted for efficient pallet loading consisting of boxes of the same or different sizes. The PLP involves many different factors, thereby making the solution NP-hard. Therefore, various methods have been developed for the solution of the PLP in the literature. In this paper, we developed a method to speed up the pallet-loading process. We utilized the three basic pallet classifications using ML. Our results provide a framework for shipment designers to decide which pallet to use when the box dimensions and demand are given. These results are significant in terms of saving considerable costs, time, and other resources.

However, our study has some limitations: it can be applied to only three of the most commonly-used standard pallet sizes. Consequently, our algorithm can be applied to only 1 in 15 real instances that satisfy these criteria.

For future research, we aim to develop an AI-based model that can efficiently predict a solution for pallet loading with variable-sized boxes. In addition, several other factors, such as moisture and PUV, will be considered. We also envision the application of neural networks to help us determine pallet sizes in the case involving more than three classes.

**Author Contributions:** B.L.A.: conceptualization, methodology, M.İ. and O.O.: software, M.İ. and O.O.: methodology, M.S.: data acquisition, writing—original draft preparation, B.L.A. and N.A.: writing—review and editing, G.S.: supervision, writing—review and editing, B.S.: funding acquisition, writing—original draft preparation. All authors have read and agreed to the published version of the manuscript.

**Funding:** This study received funding from King Saud University through researcher-supporting project number RSP-2021/145, and the APC was funded by King Saud University through researcher-supporting project number RSP-2021/145.

**Institutional Review Board Statement:** Not applicable.

**Informed Consent Statement:** Not applicable.

**Data Availability Statement:** The data used to support the findings of this study are included within the article.

**Conflicts of Interest:** The authors declare no conflict of interest.

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
