# Peer review of "Application of Machine Learning Methods for Pallet Loading Problem"

_applsci, doi:10.3390/app11188304_

Round 1

Reviewer 1 Report

The article needs to be very thoroughly revised.
There are important conceptual and methodological errors in the application of learning algorithms.

Author Response

Reviewer #1

Points

Point 1:

Reviewer 1 kindly provided a bibliography about metrics in Confusion Matrix, thus suggesting ways to deal with supervised algorithms on an imbalanced dataset. Moreover, Reviewer 1 carefully provided a list of global metrics and metrics by class that should be used to deal with the imbalanced dataset.

Response:

We thank Reviewer 1 for sharing their carefully chosen bibliography with us and we would like to point out that we took the appropriate steps to assess the predictive power of ower models with the appropriate global metrics: overall accuracy, overall balanced accuracy, Kappa or Cohen’s Kappa, Matthews Correlation Coefficient extended to multi-class, Confusion Entropy. Moreover, we used the following metrics to assess how well the five models perform on each pallet class: sensitivity by class, specificity by class, precision by class, F1 by class, balanced accuracy by class and confusion entropy by class. These metrics are defined and thoroughly explained in Section 3.3. The results are shown in Tables 2 and 3.

Point 2:

Reviewer 1 pointed out that the metrics must be calculated for the result obtained with the testing set, and not with the whole dataset.

Response:

We would like to thank Reviewer 1 for raising this crucial issue. We remedied the problem; the results being now computed for the testing set solely.

Point 3:

Reviewer 1 asked for a clarification concerning the number of instances that are used in training and testing.

Response:

We fully agree with the reviewer such a clarification is highly necessary and we now show the number of instances used in training and testing in Table 1.

Point 4:

Reviewer 1 wondered whether we have sampled according to the imbalance of the target groups when carrying out cross-validation.

Response:

Following the reviewer’s question, we have ammended our cross-validation procedure in the following way: we first split the data into 10 folds, with 9 folds being used for training and the remaining fold being used for testing. Since we are dealing with a limited amount of data and with an imbalanced data set, we used the oversampling technique called Synthetic Minority Over-sampling Technique (SMOTE) on the testing dataset in order to address the imbalanced class problem by oversampling the under-represented classes.

Point 5:

Reviewer 1 raised the issue of the necessity of including two graphics (with and without normalization) regarding the confusion matrices presented in this study.

Response:

We would like to thank the reviewer for raising this concern. Indeed, we considered that the matrices without normalization do not bring additional information and, consequently, we removed them.

Reviewer 2 Report

I think that this work gives an original point of view to the solution of the Pallet Loading Problem, taking into account modern resolution algorithm for 2D layer building and ML strategies. Moreover, it is based on real-life data, so the results are quite valuable.

However, there are some important aspects that must be clarified before the paper can be published.

  1. The authors must include a formal and precise description of the specific problem they face. For sure they consider only layered solutions (and not generic 3D packing). It seems that each layer must contain a single box type. It is not clear if a pallet must contain a single box type.
  2. Algorithm description: phase I seems to be a simple application of the selected 2D heuristics, while phase II is a trivial count if a single box type for pallet is used, while It is not clear what is done if more box types per pallet are allowed. Once the 2D layers are defied and the total layer number is selected, it is not clear what is the role of the ML algorithms. It seems to me that phase I + II already produce a solution. A possible interpretation of the ML methods is to check the quality of 2D packings, without executing the heuristics. In this case the computing time of the ML and of the heuristics is fundamental to define the best choice. Finally the authors must clarify how the overall method produces the required packing.
  3. The dataset preprocessing reduce 15187 datasets to only 1110 datasets. This seems to be done by discarding datasets that does not satisfy certain criteria. The question is: what happen to the (many) discarded data ? How can be solved the corresponding problem ? Is this method applicable only to 1 over 15 cases ?

Minor comments

  • Line 193: Dynamic compressive strength: how is it calculated? The “dynamic” term means that the compressive strength varies? I think that its definition should be reported in this work because of its importance in reference to the feasibility of pallets.
  • Lines 215-218: the notation is strange; why the 4-tple (a,b,h,w) for boxes and (L,W,H,M) for pallets ? One will expect a uniform notation as (A,B,H,W) for the pallets, or (l,w,h,m) for boxes.
  • Lines 237-238: The concept of “efficient” partition should be explained: why not < min (a,b)?
  • “Figure 4”: it seems that we have non identical boxes: squared in MB and rectangular in D
  • Confusion matrixes occupies too much space: a) remove the non-normalized matrices (do not add information); b) compact the figures in at most two rows.
  • In Figure 13: “Boxes Number”, instead of “Product No”.
  • An example of point data by dataset should be included in this paper, but it will be much better to make the datasets available for other researchers.
  • The most recent bibliography references are from 2020, and for ML related arguments.

Author Response

Reviewer #2

Major Points

Point 1:

The authors must include a formal and precise description of the specific problem they face. For sure they consider only layered solutions (and not generic 3D packing). It seems that each layer must contain a single box type. It is not clear if a pallet must contain a single box type.

Response:

Typically, PLP is divided into two categories based on the size of the loaded boxes. The first category is the uniform PLP or manufacturer’s pallet loading problem (MPLP), where the boxes are identical (homogeneous boxes). The second category of the PLP is the mixed PLP or distributor’s pallet loading problem (DPLP). The sizes of boxes are non-identical, i.e., mixed boxes of different sizes and weights (, with  being the total number of different box sizes, where  is the length of the box ,  is the width of the box ,  is the height of the box , and  is the weight of the box . The boxes can be also classified as weakly heterogeneous or strongly heterogeneous based on the sizes of the boxes in the problem. In this studies, uniform PLP is studied.

In this paper, two-phase algorithm is used to load the boxes onto the pallet. The first phase determines the number of boxes loaded onto the horizontal layer. However, one of the main assumptions is to   ensure that the boxes fit completely on the pallet without any overlap and overhang. If the pallet area divided by the box area is < 101 then researchers (i.e. Martins, & Dell, 2008) show that either one-block heuristic, three-block heuristic, five-block heuristic, hollow-block heuristic or the G5-heuristic could obtain the maximum number of identical boxes per a horizontal layer (Bischoff, Janetz, & Ratcliff, 1995; Steudel, 1979; Liu, & Hsiao, 1997; Martins, & Dell, 2007, 2008, Nelissen, 1993). The second phase, the number of horizontal layers loaded onto the pallet is calculated based on the max height (, maximum weight (, and the dynamic compression strength considering humidity, interlock stacking pattern, and storage time. However, this study was develop based on the real-life problems from the DHL Supply Chain.

Point 2:

Algorithm description: phase I seems to be a simple application of the selected 2D heuristics, while phase II is a trivial count if a single box type for pallet is used, while it is not clear what is done if more box types per pallet are allowed. Once the 2D layers are defied and the total layer number is selected, it is not clear what is the role of the ML algorithms. It seems to me that phase I + II already produce a solution. A possible interpretation of the ML methods is to check the quality of 2D packings, without executing the heuristics. In this case the computing time of the ML and of the heuristics is fundamental to define the best choice. Finally, the authors must clarify how the overall method produces the required packing.

Response:

The pallet loading problem is one of the main problems that has been deeply studied in the literature. As of yet, there was no studies covered the pallet loading problem considering the maximum height, maximum weight, and dynamic strength. In this study we aimed to propose a new two-algorithm to load the identical boxes onto the pallet considering the effect of temperature and the humidity. Aforementioned, the uniform pallet loading problem is one of the problems in the manufacturing systems, warehouses, and distribution centers.

ML is utilized to determine pallet size. The input necessary to solve this problem comes from multiple layout heuristics where layout is determined, and efficiency values are determined. Later this is fed into ML to select the pallet size.

Point 3:

The dataset preprocessing reduces 15187 datasets to only 1110 datasets. This seems to be done by discarding datasets that does not satisfy certain criteria. The question is: what happen to the (many) discarded data? How can be solved the corresponding problem? Is this method applicable only to 1 over 15 cases?

Response:

We thank reviewer 2 for their question.  Indeed, this method is applicable at the moment only in 1 over 15 cases. Future extensions to this work are being considered, namely we envision the application of Neural Networks to help us determine pallet sizes in the case where we are dealing with more than three classes.

Minor Comments

Point 1:

Line 193: Dynamic compressive strength: how is it calculated? The “dynamic” term means that the compressive strength varies? I think that its definition should be reported in this work because of its importance in reference to the feasibility of pallets.

Response:

We think the reviewer mentioned a very worthy point in this context. As such, we are considering it. Per reviewer recommendations, subsection 3.1.3 has been modified in the revised revision to explain the dynamic strength term.

Point 2:

Lines 215-218: the notation is strange; why the 4-tple (a,b,h,w) for boxes and (L,W,H,M) for pallets ? One will expect a uniform notation as (A,B,H,W) for the pallets, or (l,w,h,m) for boxes.

Response:

The notations have been changed for consistency purposes, where L, W, H and are used to represent the pallet dimensions and l, w, h, and m are utilized to refer to the box dimensions. 

 Point 3:

Lines 237-238: The concept of “efficient” partition should be explained: why not < min (a,b)?

Response:

Efficient partition is used to ensure that the boxes fit completely within the boundaries of the pallet.

Point 4:

“Figure 4”: it seems that we have non identical boxes: squared in MB and rectangular in D

Response:

Changed

Point 5:

Confusion matrixes occupies too much space: a) remove the non-normalized matrices (do not add information); b) compact the figures in at most two rows.

Response:

We would like to thank the reviewer for raising this concern. Indeed, we considered that the matrices without normalization do not bring additional information and, consequently, we removed them.

Point 6:

In Figure 13: “Boxes Number”, instead of “Product No”.

Response:

Changed

Point 7:

An example of point data by dataset should be included in this paper, but it will be much better to make the datasets available for other researchers.

Response:

Our co-author from DHL provided the data. The data is not publicly available. DHL is a big company, and they handle a large number of transactions daily. If the company permits, we do not mind sharing the data ourselves.

Point 8:

The most recent bibliography references are from 2020, and for ML related arguments.

Response:

Done

Reviewer 3 Report

The article presents the results of research on the application of several machine learning algorithms to solve simple pallet loading tasks. The advantage of the work was access to a large set of real data. Many simplifying assumptions were made in the research. This applies in particular to the homogeneity of loading units (containers) on pallets, arranged in a single layer, according to the same pattern on a pallet.

In the research, the emphasis was placed on the speed of obtaining solutions. The quality (accuracy) of the results in relation to the solutions obtainable with other methods (strict, heuristic, etc.) was not commented on.

In the introduction and in the end, I propose to refer to this issue, as well as to the possibility of including in the algorithms several important limitations from the point of view of practice, such as:

- arranging different (with different dimensions) containers on a pallet,

- multi-layer loading of pallets and the compressive strength of loads as well as the lateral stability of such piled loads

- transport (and storage) conditions, including humidity, transport, and storage time (such declaration appeared in the introduction).

Author Response

Reviewer #3

Comments

Point 1:

The Reviewer shared his/her concern that in the research the emphasis was placed on the speed of obtaining solutions and that the accuracy of the results in relation to the solutions obtainable with other methods (strict, heuristic) was not commented on.

Response:

We thank reviewer 3 for raising this crucial issue and we would like to highlight the fact that a thorough review of previous research has been presented in Section 2 of our manuscript. For instance, we covered the fact that solutions based on classical mathematical models conduct an exhaustive research of all the possibilities which ideally obtain 100% accuracy are impractical for large problem sizes. The use of heuristics was also covered, and we underscored the importance of Martin and Dell’s study which was able to determine optimal solutions for 99.7% of problems. Following reviewer 3’s suggestion, we also highlighted the high accuracy of our Machine Learning models, in addition to emphasizing their speed, the latter becoming a significant factor when dealing with large-scale problems.

Point 2:

Arranging different (with different dimensions) containers on a pallet.

Response:

Thank you so much for your valuable comment. We are planning to study this is in detail in future works.

Point 3:

Multi-layer loading of pallets and the compressive strength of loads as well as the lateral stability of such piled loads

Response:

There are two patterns of stacking, when one or more type of horizontal layer is utilized to load the boxes per pallet. The first pattern is columnar stacking, where all horizontal layers in the pallet have the same layout and pattern. In this pattern, the boxes are loaded into horizontal layers, wherein the boxes are edge-to-edge and corner-to-corner, as well as all horizontal layers have the same pattern, and they belong to the same type of horizontal layer. The strength of boxes is high, while the stiffness and stability are low due to the load distribution at a certain point. From a strength perspective, the compression strength, where 2/3 of potential compression exists in the vertical corners and edges, is the main aspect taken into consideration in literature. In this regard, the boxes should be seamed edge-to-edge and corner-to-corner for the greatest stacking strength. The second pattern is the interlock stacking pattern. This pattern appears in the pallet when several types of horizontal layers are utilized to load the boxes onto a pallet, or if the altered layouts of the same horizontal layer are used to load the boxes onto pallet. In this pattern, the edge-to-edge and corner-to-corner are not matched; therefore, it affects the box strength. The researchers show that the interlocking pattern reduces the dynamic strength of boxes. Meanwhile it improves the stability and stiffness of boxes and pallet compared with columnar stacking pattern.

This study develops based on the first pattern, columnar stacking, to improve the strength of the boxes. Besides this study, we are working on another paper, where it assumes that using the interlocking pattern among boxes in the pallet to create a more stable pallet, once more than one pallet can be stacked together. Also, more than one type of horizontal layer is utilized to load the boxes onto a pallet, as a result the interlock stacking pattern appears among boxes. In this regard, the load distribution across the pallet is uniform when the interlock stacking pattern is utilized to load the boxes onto a pallet.

Point 4:

Transport (and storage) conditions, including humidity, transport, and storage time (such declaration appeared in the introduction).

Response:

We think the reviewer mentioned a very worthy point in this context. As such, we are considering it. Per reviewer recommendations, subsection 3.1.3 has been modified in the revised revision to explain the dynamic strength term.

Round 2

Reviewer 1 Report

Yes, you are now properly using the ML and confusion matrix.

Please review the style and some orthography mistakes (i.e., repeated contiguous words). Now, the methodology is correct and I would say that the paper is formally correct.

Author Response

Reviewer #1

Minor Comments

Point 1:

Reviewer 1 pointed out that please review the style and some orthography mistakes (i.e., repeated contiguous words).

Response:

We thank Reviewer 1 for sharing this issue. We corrected all of them. Moreover, the format was revised as per journal guidelines.

Reviewer 2 Report

The authors performed a wide revision addressing the majority of my concerns.

There is one minor point that is clearly stated in the response, but not in the paper: The authors consider a benchmark set of 15K instances and reduces to 1K before applying their method. In the response they observe that their method can be applied only to instances with specific requirements, so just 1 over 15 can be solved. This is something that must be clearly written in the paper: both the limiting hypothesis for the application of the algorithm and the fact that just 1 over 15 real instances satisfies such hypothesis. Consequently the figures and descriptions where it is presented a process that reduce 15K instances to 1K must be updated by removing this (misleading) reduction, but simply stating that you consider the 1K instances for which the algorithm can be applied.

Moreover, I will see appropriate to cite some more papers from MDPI. Here some examples.

Complexity Constraint in the Distributor’s Pallet Loading Problem, by Hugo Barros, Teresa Pereira,  António G. Ramos, and Fernanda A. Ferreira,  Mathematics 2021, 9(15), 1742; https://doi.org/10.3390/math9151742

Solving a Real-Life Distributor’s Pallet Loading Problem, by Mauro Dell’Amico and Matteo Magnani, Math. Comput. Appl. 2021, 26(3), 53; https://doi.org/10.3390/mca26030053

A Two-Phase Approach for Single Container Loading with Weakly Heterogeneous Boxes, by Rommel Dias Saraiva, Napoleão Nepomuceno and Plácido Rogério. Algorithms 2019, 12(4), 67; https://doi.org/10.3390/a12040067

Study of the Stability of Palletized Cargo by Dynamic Test Method Performed on Laboratory Test Bench, by Sławomir Tkaczyk, Mikołaj Drozd, Łukasz Kędzierski and Krzysztof Santarek, Sensors 2021, 21(15), 5129; https://doi.org/10.3390/s21155129

Author Response

Reviewer #2

Minor Comments

Point 1:

Reviewer 2 raised this following crucial point: The authors consider a benchmark set of 15K instances and reduces to 1K before applying their method. In the response they observe that their method can be applied only to instances with specific requirements, so just 1 over 15 can be solved. This is something that must be clearly written in the paper: both the limiting hypothesis for the application of the algorithm and the fact that just 1 over 15 real instances satisfies such hypothesis. Consequently the figures and descriptions where it is presented a process that reduce 15K instances to 1K must be updated by removing this (misleading) reduction, but simply stating that you consider the 1K instances for which the algorithm can be applied.

Response :

We are thankful for Reviewer 2’s contribution to the clarification of this important issue. We now clearly state in the Dataset section of our article that our algorithm is applicable only for  the P1, P2, and P3 pallet sizes, with only approximately 1 in 15 instances being further considered. Next, we updated one paragraph in the Conclusions section underlining the limitations of our study. Moreover, for the figures in which the P1, P2, and P3 classes are presented we highlighted that these represent the 1K data points for which the algorithm could be applied.

Point 2:

Reviewer 2 carefully provided a bibliography of latest research articles that deal with the complexity of PLP which should be referenced in our study.

Response :

We thank Reviewer 2 for the selected references, which have been incorporated into our research paper as an extra paragraph in the introduction. We feel that these additions enriched our article and provided a more nuanced overview of the complexity of the PLP problem.
